# RecLM: Recommendation Instruction Tuning with Large Language Models

## Abstract

Recommender systems aim to deeply understand users' complex preferences based on their past interactions. Deep collaborative filtering paradigms, leveraging advanced neural architectures like Graph Neural Networks (GNNs), excel at capturing collaborative relationships among users. However, limitations emerge when dealing with sparse data or zero-shot learning from unseen datasets, due to the design constraints of ID-based embedding functions in existing solutions. These challenges hinder robust generalization and adaptability. To address this, we propose a model-agnostic recommendation instruction-tuning paradigm that integrates large language models with collaborative filtering. Our Recommendation Language Model (RecLM) is introduced to enhance the capability of capturing user preference diversity. We design a reinforcement learning reward function to facilitate self-augmentation of our language models. Comprehensive evaluations demonstrate significant advantages of our approach across various settings. It can be integrated as a plug-and-play component with state-of-the-art recommender systems, resulting in notable performance enhancements. We have made our RecLM available anonymously at: https://anonymous.4open.science/r/RecLM-A1BE/.

## 1 Introduction

Recommendation systems play a vital role in web applications, assisting users in navigating the vast amount of information accessible online. These systems deliver personalized recommendations of items that users may find interesting, including products on e-commerce platforms Wang et al. (a); Wu et al., posts on social networking sites Jamali & Ester; Zhang et al. (2021), and videos on sharing platforms Wei et al. (a); Zhan et al.. One of the most widely used methods for generating these recommendations is Collaborative Filtering (CF). This approach leverages the preferences of similar users to suggest new items to a specific user He et al. (2017).

However, it is essential to emphasize that most of current recommender systems primarily rely on the user/item ID paradigm, where the training data largely consists of mapped user and item indices. While this approach has significantly advanced recommendation, particularly in scenarios with ample training data Yuan et al., it also presents notable limitations. Key challenges include suboptimal performance in cold-start scenarios and difficulties in generalizing to zero-shot learning situations. In completely cold-start settings, ID-based recommenders struggle to generate effective representations for new items, often leading to failures in providing valid recommendations.

To address the cold-start challenge in the ID-based recommendation paradigm, a promising approach is to leverage external features (*e.g.*, textual or visual information) associated with users and items to generate their representations, rather than relying on ID-based embeddings. However, real-world scenarios often lack complete modal features. For instance, many users may withhold personal information due to privacy concerns, resulting in incomplete data. Additionally, these external features often contain noise, which can distort modeling of user preferences. For example, misleading tags or inaccurate specifications in an item's description may lead to misguided recommendations. Consequently, extracting accurate, relevant, and high-quality external features from noisy and incomplete data has become a critical challenge for generalizing recommenders under data scarcity.

**Contribution.** Inspired by the robust generalization and reasoning capabilities of Large Language Models (LLMs), we propose the development of effective language models as profiling systems specifically designed for recommendation tasks, aimed at enhancing performance in cold-start

recommendation scenarios. Utilizing LLMs for profile generation involves addressing two primary challenges: (i) How can LLMs generate profile text that accurately reflects the recommendation characteristics for users or items lacking external features? (ii) How can LLMs produce high-quality profiles from noisy features while effectively capturing user-item interaction behavior context?

To address these challenges, we propose a novel approach that involves performing message passing among users and items to enrich user and item profiling with information from their interactions. This method allows users and items with insufficient external features to be effectively profiled through their interaction dependencies from a global perspective. Additionally, we introduce an innovative recommendation instruction tuning paradigm that integrates behavioral signals into LLMs. This paradigm enables LLMs to not only incorporate external features from users and items but also to understand user preferences in the context of user-item interaction data. By guiding LLMs to consider collaborative relationships, this approach addresses the lack of direct supervision signals in profile generation tasks through self-supervised learning. Furthermore, to mitigate the extraneous noise introduced by this instruction tuning paradigm and counteract the over-smoothing caused by collaborative relationships, we propose a reinforcement learning-based personalized feature enhancement method. This technique aims to further improve the accuracy and personalization of the generated profiles. Our main contributions can be summarized as follows:

- **Model-Agnostic Framework**. We introduce a model-agnostic instruction tuning framework RecLM. It can be seamlessly integrated into existing recommender systems as a plug-and-play component, significantly enhancing their generalization capacity in scenarios with limited data.
- **Enhancing Profiling System**. In this work, we seamlessly integrate large language models with collaborative filtering to enhance user profiling, particularly in cold-start scenarios, where current methods often struggle. Additionally, our approach employs reinforcement learning to refine profile quality, effectively addressing challenges associated with data noise and over-smoothing.
- **Comprehensive Evaluation**. We integrate RecLM with a range of state-of-the-art recommenders to assess the effectiveness of our approach across various settings. This includes conducting ablation studies and efficiency evaluations. Additionally, we carry out extensive experiments in real-world industrial recommendation scenarios, demonstrating the practicality and scalability of RecLM.

## 2 RELATED WORK

### 2.1 ID-BASED RECOMMENDER SYSTEMS

In recommender systems, numerous collaborative filtering models have been proposed to map users and items into latent representations based on user/item IDs Koren et al. (2021); Su & Khoshgoftaar (2009). These methods have evolved significantly, starting from early matrix factorization techniques, such as BiasMF Koren et al. (2009), to the introduction of Neural Collaborative Filtering (NCF) with the advent of neural networks He et al. (2017). Recently, advancements in Graph Neural Networks (GNNs) have opened promising avenues for constructing bipartite graphs based on user-item interaction history, allowing for the capture of high-order collaborative relationships. GNN-based methods, including NGCF Wang et al. (2019), GCCF Chen et al., and LightGCN He et al. (2020), have demonstrated state-of-the-art performance, enhancing the effectiveness of recommendation.

Additionally, researchers have incorporated self-supervised learning (SSL) techniques as supplementary learning objectives to improve the robustness of recommenders and address challenges related to data sparsity and noise Yu et al. (2023). Contrastive learning (CL), a widely adopted SSL technique, has been effectively applied in CF research through approaches such as SGL Wu et al. (2021), SimGCL Yu et al. (2022), NCL Lin et al., and AdaGCL Jiang et al.. Despite these advancements, ID-based recommenders still face significant limitations, particularly in completely cold-start scenarios and in terms of model transferability Yuan et al..

### 2.2 LARGE LANGUAGE MODELS (LLMS) FOR RECOMMENDATION

The application of large language models (LLMs) in recommender systems has garnered significant attention Fan et al. (2023); Lin et al. (2023); Liu et al. (2023). Current approaches can be categorized into two main types. The first category includes methods such as P5 Geng et al. and Chat-REC Gao et al. (2023), which emphasize designing prompts aligned with recommendation tasks, utilizing the

LLM directly as the inference model. The second category enhances existing recommenders by integrating LLMs while still relying on traditional collaborative filtering methods Wang et al. (b). For instance, LLMRec Wei et al. (b) strengthens the user-item interaction graph through LLM-based graph augmentation, while RLMRec Ren et al. (2023) combines LLM-enhanced text embeddings with GNN-based user/item representations. However, these approaches often lack fine-tuning tailored to specific recommendation tasks, primarily focusing on full-shot scenarios.

In contrast, our work introduces a novel instruction-tuning technique for an open-source LLM, allowing it to adapt to specific recommendation tasks and effectively capture collaborative information for profile generation. While methods like InstructRec Zhang et al. (2023) and TALLRec Bao et al. align LLM capabilities with recommendation tasks, they struggle with scalability due to instruction-question-answering prompts and exhibit poor generalization on sparse data. Our approach enhances the generalization ability of existing recommender systems in the face of data scarcity and noise, while maintaining efficiency in handling large-scale data in practical scenarios.

## 3 METHODOLOGY

### 3.1 ID-BASED COLLABORATIVE FILTERING

In the ID-based collaborative filtering (CF) paradigm, the main goal is to optimize the ID embeddings of users and items. This optimization aims to accurately capture and represent user preferences for items, while considering the interaction patterns of users and items that are similar. Formally, we have a set of users denoted as $\mathcal{U} = \{u_1, \cdots, u_I\}$, and a set of items denoted as $\mathcal{V} = \{v_1, \cdots, v_J\}$. Each user and item is assigned initial ID embeddings, represented as $\mathbf{x}_u$ and $\mathbf{x}_v \in \mathbb{R}^d$ respectively. The objective is to obtain optimized user and item representations, denoted as $\mathbf{e}_u, \mathbf{e}_v \in \mathbb{R}^d$, through a recommender model $\mathcal{R}(\mathbf{x}_u, \mathbf{x}_v)$. This model aims to maximize the posterior distribution $p(\mathbf{e}|\mathcal{X}) \propto p(\mathcal{X}|\mathbf{e})p(\mathbf{e})$. The predicted likelihood of user-item interaction, denoted as $\hat{y}_{u,v}$, is derived by performing a dot product between the user and item representations, as follows: $\hat{y}_{u,v} = \mathbf{e}_u^\top \cdot \mathbf{e}_v$.

Although many state-of-the-art recommender systems operating within the ID-based collaborative filtering paradigm have demonstrated remarkable performance, they face significant challenges when it comes to handling item cold-start scenarios, especially in situations where data scarcity is prevalent. The primary hurdle arises from the lack of past interaction history for these new items, which disrupts the optimization paradigm mentioned earlier. As a consequence, ID-based recommenders may encounter difficulties in generating accurate representations for these new items, leading to a notable decline in the overall performance of recommender systems, particularly in zero-shot scenarios.

### 3.2 TEXT-EMPOWERED USER/ITEM REPRESENTATIONS

To handle cold-start items in zero-shot recommendation, we propose to leverage textual side features for user and item representation learning. Specifically, we propose to replace the aforementioned ID embeddings with the side information associated with items, concretely items' text descriptions $\mathbf{F} \in \mathbb{R}^{|\mathcal{V}| \times d_t}$. A multi-layer perceptron $T_{raw}$ is utilized to project the raw textual features $\mathbf{f} \in \mathbb{R}^{d_t}$ into the latent space $\mathbb{R}^d$. The resulting representation $\hat{\mathbf{f}} \in \mathbb{R}^d$ is then used for initial item representation:

$$\hat{\mathbf{f}}_v = T_{raw}(\mathbf{f}). \tag{1}$$

This enables items to have meaningful representations with textual semantics that go beyond simple ID embeddings. After using textual features as item representations, the recommender system optimizes the user ID embedding using observed item interactions, capturing user preferences for text-based items and enabling zero-shot predictive capabilities for cold-start items.

**LLM-enhanced User/Item Profiling**. To further empower user representations with the rich textual semantics provided by large language models (LLMs), we propose generating user profile information that can reflect their interaction preferences. Specifically, item profiles can be derived from the profiles of users who frequently interact with them. This approach proves valuable in capturing user preferences and facilitating accurate recommendations for cold-start items. On the user side, the original ID embedding $\mathbf{x}_u \in \mathbb{R}^d$ is seamlessly integrated with the user profile representation $\mathbf{p}_u \in \mathbb{R}^{d_t}$, allowing the system to leverage both the user's ID-based embedding and their generated profile representation, which can capture more nuanced preferences. Similarly, on the item side, the

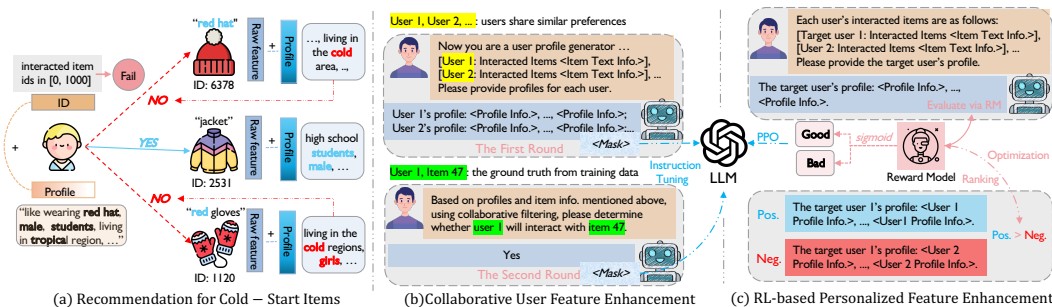

Figure 1: Overall framework of the proposed RecLM.

raw text features of the item $\mathbf{x}_v$ are effectively combined with the item profile $\mathbf{p}_v \in \mathbb{R}^{d_t}$, enabling the system to better understand the item's characteristics and how they align with user preferences.

$$\hat{\mathbf{f}}_u^{aug} = \Psi(\mathbf{x}_u \mid T_{pro}(\mathbf{p}_u)), \quad \hat{\mathbf{f}}_v^{aug} = \Psi(\hat{\mathbf{f}}_v \mid T_{pro}(\mathbf{p}_v)). \tag{2}$$

To fuse the multi-faceted information, we employ an MLP, $\Psi$, to consolidate the various features. Additionally, we use another MLP, $T_{pro}$, to convert the profile embeddings into the model's latent space. This fusion process produces enhanced user and item representations, $\hat{\mathbf{f}}_u^{aug} \in \mathbb{R}^d$ and $\hat{\mathbf{f}}_v^{aug} \in \mathbb{R}^d$, which prove instrumental in accurately predicting user behavior. Given the rapid development and widespread use of LLMs, their data augmentation capabilities have showcased impressive performance. Leveraging this power, we utilize LLMs to generate supplementary profiles for users and items, effectively boosting the capabilities of our recommender system.

### 3.3 ENHANCING COLLABORATIVE FEATURES VIA RECOMMENDATION INSTRUCTION TUNING

To enhance collaborative features, our RecLM proposes to integrates users' collaborative relationships into the aforementioned LLM-based profiling process, through an innovative recommendation instruction tuning paradigm. This approach improves the generated user profiles by employing knowledge distillation and a dialogue-based instruction tuning method, effectively preserving high-order collaborative similarities between users and items. Once we have successfully generated high-quality user profiles, we can proceed to generate item profiles by leveraging the associated user profiles, ensuring semantic alignment and resulting in enhanced features for both users and items.

#### 3.3.1 **LLM Fine-Tuning via Knowledge Distillation**

For our profiling system, using state-of-the-art LLMs like ChatGPT can be costly and inefficient, with data security concerns. Instead, fine-tuning open-source LLMs is more common, granting flexibility to align with computational resources and business needs. This allows designing cost-effective and efficient batch inference methods while ensuring data security. Here, we utilize llama2-7b-chat as the base model. To tailor it to our business, we design prompt templates and sample users and items to construct input prompts for ChatGPT-3.5. After obtaining the inference results, we fine-tune llama2-7b-chat using the input-output prompts. This distills knowledge from the large-scale ChatGPT into the open-source llama2-7b-chat, yielding a fine-tuned LLM $\mathcal{M}_{kd}$ that meets our requirements.

#### 3.3.2 COLLABORATIVE INSTRUCTION TUNING

Indeed, the reliance solely on historical item information from user interactions may not effectively harness the collaborative relationships among users. Therefore, we have devised a solution to this issue by introducing a two-turn dialogue-based instruction tuning paradigm. This paradigm not only aids LLMs in generating higher-quality profiles by considering the collaborative relationships among users but also tackles the challenge of lacking direct supervision in the profile generation task.

**Profile Generation with Two-turn Dialogue.** The challenge of evaluating generated profiles without readily available ground truth hinders the guidance of LLMs in producing high-quality profiles. Typically, the quality assessment is indirectly conducted through downstream recommendation tasks

where the profiles are utilized. To tackle this challenge, we have devised a two-turn dialogue-based instruction-tuning paradigm.

In the first turn, the input $query$ (*i.e.,* $\mathcal{Q}$) encompasses the historical item lists of the target user and similar users. The output $response$ (*i.e.,* $\mathcal{R}$) is the profiles of users. Our aim in this turn is guiding LLMs to consider both the collaborative relationships among users and their historical interactions when generating the user's profile. However, the profile generated in the first turn is solely based on the distilled knowledge of $\mathcal{M}_{kd}$ and lacks sufficient supervision signals for effective guidance.

To address this, we introduce the second turn where we utilize the user's historical interaction records as supervision signals to guide the profile generation process. In this second turn, the input $\mathcal{Q}$ consists the target user $u_t$ and a target item $v_t$. The output $\mathcal{R}$ is a prediction of whether $u_t$ will interact with $v_t$. This approach bridges the gap between the profile generation task and the recommendation optimization objective, guiding LLMs to consider the collaborative relationships among users during profile generation and allows for supervision signals from the recommendation task.

**Instruction Design.** In the first turn, LLMs receive the historical item list $\mathcal{V}_t \in \mathcal{V}$ of the target user $u_t \in \mathcal{U}$ and the historical item lists $\mathcal{V}_n \in \mathcal{V}$ of several users $u_n \in \mathcal{U}$ with similar preferences. To identify these similar users $\{u_n\}$, we employ traditional ID-based recommenders to obtain user embeddings. By calculating the cosine similarity between user embeddings, we can obtain users who exhibit comparable preferences. Alongside $\mathcal{V}_n$, the LLM is also provided with the user profiles of these similar users via $\mathcal{M}_{kd}$. The output $\mathcal{R}$ of LLMs in this turn is the profiles of both $u_t$ and $\{u_n\}$.

$$\mathcal{Q}_{fir.} = Prompt(u_t, \{u_n\}, \mathcal{V}_t, \{\mathcal{V}_n\}), \quad \mathcal{R}_{fir.} = Prompt(u_t, \{u_n\}, \mathcal{P}_t, \{\mathcal{P}_n\}). \tag{3}$$

In the second turn, the input $\mathcal{Q}$ revolves around whether $u_t$ will interact with $v_t$. The output $\mathcal{R}$ indicates the interaction status between $u_t$ and $v_t$ in the training dataset (*i.e.,* $yes$ or $no$). To maintain a balanced distribution of positive and negative samples, we employ the following approach: For half of the samples (*i.e.,* positive samples), $v^+$ is chosen from $\mathcal{V}_t$. Additionally, it is ensured that $v^+$ appears in the interaction history $\{\mathcal{V}_n\}$. Meanwhile, when constructing the instructions for the first turn of the dialogue, $v^+$ is removed from $\mathcal{V}_t$. For the remaining half of the samples (*i.e.,* negative samples), $v^-$ is selected from $\{\mathcal{V}_n\}$. Importantly, $v^-$ has not been interacted with by $u_t$ in the training dataset. This approach ensures that the dialogue instructions maintain a balance between positive and negative samples, while also incorporating relevant contextual information without introducing bias towards any specific item.

$$\mathcal{Q}_{sec.} = \begin{cases} Prompt(u_t, v^+), & pos.\ samp. \\ Prompt(u_t, v^-), & neg.\ samp. \end{cases} \quad \mathcal{R}_{sec.} = \begin{cases} \text{Yes,} & pos.\ samp. \\ \text{No,} & neg.\ samp. \end{cases} \tag{4}$$

**Tuning Strategy.** In the process of multi-turn dialogue instruction-tuning, the object is to utilize the $\mathcal{R}$ generated by LLMs for weight updates, while excluding the $\mathcal{Q}$ from these updates. If simply employing the conventional single-turn dialogue tuning approach on our paradigm, the inputs to the LLM include $\mathcal{Q}_{fir.}$, $\mathcal{R}_{fir.}$, and $\mathcal{Q}_{sec.}$, with only $\mathcal{R}_{sec.}$ being the predicted part. Therefore, only the loss from $\mathcal{R}_{sec.}$ is utilized for updating LLM's weights, which fails to fully exploit the training data for multi-turn dialogues. In our designed paradigm, $\mathcal{R}_{fir.}$ contains valuable textual information in the form of profiles of multiple users. This rich information guides the generation of $\mathcal{R}_{sec.}$ in the subsequent turn. On the other hand, $\mathcal{R}_{sec.}$ is the relatively simple text, usually a binary choice such as $yes$ or $no$. If we disregard the information in $\mathcal{R}_{fir.}$ and solely use $\mathcal{R}_{sec.}$ for fine-tuning LLMs, it is evident that we would not be able to achieve the desired effect.

To address this issue, we have devised a more efficient method for two-turn dialogues tuning. Our approach involves concatenating the data from the two-turn dialogues and utilizing masking techniques to distinguish between $\mathcal{Q}$ and $\mathcal{R}$. When updating the weights of LLMs, only the loss from the part marked as $\mathcal{R}$ is taken into account for weight updates. By adopting this method, both $\mathcal{R}_{fir.}$ and $\mathcal{R}_{sec.}$ in the two-turn dialogue are able to actively contribute to the training process, allowing for the full utilization of the dialogue data. This approach is instrumental in guiding LLMs within our designed paradigm to effectively learn the collaborative relationships among users.

**Inference Prompt.** After completion of the instruction-tuning, we have devised a prompt for inferring user profiles. This prompt combines both $\mathcal{V}_t$ and $\{\mathcal{V}_n\}$. Its purpose is to provide guidance to LLMs

in generating user profiles that are enriched through collaborative relationships among users.

$$\mathcal{Q}_{inf.} = Prompt(u_t, \{u_n\}, \mathcal{V}_t, \{\mathcal{V}_n\}). \tag{5}$$

## 3.4 REINFORCEMENT LEARNING-BASED PERSONALIZED FEATURE ENHANCEMENT

Despite the LLM's improved ability to infer profiles through the instruction-tuning, there are still challenges to address. The inconsistency between the prompt instruction during inference (*i.e.,* generates a specific user's profile) and the first turn of the dialogue during fine-tuning (*i.e.,* generates multiple users' profiles) introduces noise into the generated profiles. Additionally, while considering user collaborative relationships enhances performance, it compromises profile precision in terms of personalization, similar to the over smoothing problem in GNN-based CF methods.

To address above challenges, we draw inspiration from the RLHF (Reinforcement Learning from Human Feedback) technique Stiennon et al., and develop a RL-based fine-tuning paradigm to further enhance the previously tuned LLM. In this approach, we train a reward model to evaluate the quality of the profiles generated by the LLM. Subsequently, we employ the Proximal Policy Optimization (PPO) Schulman et al. (2017) to update the weights of the LLM using the scores provided by the reward model. This iterative process enables us to progressively refine the LLM's performance, resulting in the generation of more accurate and personalized profiles.

**Reward model.** The goal of a reward model is to characterize whether the LLM's output is considered good by humans. That is, given an input pair of $[\mathcal{Q}, \mathcal{R}]$, it outputs a scalar value that represents the quality of the $\mathcal{R}$. The optimization loss $\mathcal{L}_{rm}$ for the reward model is as follows:

$$\mathcal{L}_{rm} = -\sum_{i=0}^{\mathcal{I}} \mathbb{E}_{(\mathcal{Q}_i, \mathcal{R}_i^+, \mathcal{R}_i^-) \sim D}[\log(\sigma(r_\theta(\mathcal{Q}_i, \mathcal{R}_i^+) - r_\theta(\mathcal{Q}_i, \mathcal{R}_i^-)))], \tag{6}$$

where $r_\theta(\cdot)$ denotes the reward model, $\sigma(\cdot)$ denotes $sigmod$ function, $\mathcal{R}_i^+$ and $\mathcal{R}_i^-$ are true response and false response respectively. The success of training the reward model relies on high-quality and effective training data. In the context of the profiling task, we utilize the same query (*i.e.,* $\mathcal{Q}_{inf.}$). The critical aspect is to construct both *positive* response (*i.e.,* $\mathcal{R}^+$) and *negative* response (*i.e.,* $\mathcal{R}^-$). For $\mathcal{R}^+$, we obtain profiles via ChatGPT. As for $\mathcal{R}^-$, we categorize them into two groups. Firstly, we design multiple prompt templates to generate diverse negative samples. These samples assist the reward model in learning to distinguish low-quality responses that the LLM may generate after the previous instruction-tuning. Secondly, we substitute the target user's profile with profiles of similar users. This aids the reward model in discerning between similar profiles and selecting more personalized and accurate ones. By incorporating these techniques, we enhance the training data and improve the reward model's ability to evaluate the quality of generated user profiles.

**Proximal Policy Optimization.** Following the conventional RL framework, where the reward model serves as an approximation of the true reward function, the LLM $\mathcal{M}$ is treated as the policy to be optimized. The optimization objective in this process is as follows:

$$\underset{\mathcal{M}}{\text{argmax}} \, \mathbb{E}_{x_i \sim \mathcal{D}, y_i \sim \mathcal{M}}[R(y_i|x_i)]. \tag{7}$$

To iteratively optimize $\mathcal{M}$, we sample $\mathcal{Q}_i$ from the query set $\mathcal{D}$ and the corresponding $\mathcal{R}_i$ generated via $\mathcal{M}$. We utilize the Proximal Policy Optimization (PPO) algorithm and its associated loss function to achieve this objective. Following Schulman et al. (2017), the final reward function contains an additional penalty term (*i.e.,* KL divergence of the original LLM $\mathcal{M}_0$ and the optimizing LLM $\mathcal{M}_\theta$). This constraint is beneficial to reducing reward hacking whereby achieving high scores from the reward model but low scores from real human evaluation. Hence, the final reward function $R(\cdot)$ for the sample $\mathcal{R}_i$ and $\mathcal{Q}_i$ is as follows:

$$R(\mathcal{R}_i|\mathcal{Q}_i) = \hat{r}(\mathcal{R}_i|\mathcal{Q}_i) - \beta D_{KL}(\mathcal{M}_\theta(\mathcal{Q}_i)||\mathcal{M}_0(\mathcal{Q}_i)) \tag{8}$$

We report the detailed instruction designs for fine-tuning at each stage of our work, along with the construction of positive and negative training samples for the RL reward model in Appendix A.6.

## 4 EVALUATION

In this section, we verify the effectiveness of RecLM by answering the following several questions:

- **RQ1**: How does our proposed RecLM enhance the performance of existing recommender systems, particularly in item cold-start scenarios?

- **RQ2**: What contributions do the instruction-tuning techniques and reinforcement learning enhancements make to overall recommendation performance?

- **RQ3**: How effective is our LLM-empowered user/item profiling system as an embedding function?

- **RQ4**: How does our method perform in terms of efficiency?

- **RQ5**: What advantages does our method have compared to existing LLM-enhanced recommenders?

- **RQ6**: How does the reinforcement learning-based feature enhancement module enhance the performance of our LLM-empowered profiling system?

## 4.1 EXPERIMENTAL SETTINGS

To evaluate the effectiveness of our proposed method, we conduct extensive experiments using two public datasets: **MIND**[1] Wu et al. (2020) and **Netflix**[2], along with a large-scale dataset derived from real-world industrial data (referred to as Industrial for anonymity).

We assess the accuracy of the top-*K* recommendation results using two widely adopted metrics: Recall@K (R@K) and NDCG (N@K), with $K$ set to 20 by default. To reduce bias, we employ an all-rank evaluation strategy, where positive items in the test set are ranked alongside all non-interacted items for each user. The final metric is reported as the average score across all users in the test set.

We evaluate the effectiveness of our RecLM approach by integrating it with state-of-the-art recommender systems, allowing us to assess performance improvements in a model-agnostic manner compared to baseline models. The selected CF recommenders include non-graph methods such as BiasMF Koren et al. (2009) and NCF He et al. (2017), the GNN-enhanced method LightGCN He et al. (2020), and graph contrastive learning approaches SGL Wu et al. (2021) and SimGCL Yu et al. (2022). Details regarding the datasets and baseline methods are provided in Appendices A.1 and A.2.

## 4.2 PERFORMANCE COMPARISON (RQ1)

Table 1: Performance comparison on MIND, Netflix and Industrial data in terms of *Recall* and *NDCG*. The superscript * indicates the improvement is statistically significant where the p-value $< 0.05$.

| Dataset | | MIND | | | | Netflix | | | | Industrial | | | |
|---|---|---|---|---|---|---|---|---|---|---|---|---|---|
| Backbone | Variants | R@20 | R@40 | N@20 | N@40 | R@20 | R@40 | N@20 | N@40 | R@20 | R@40 | N@20 | N@40 |
| | | | | | | Full-Shot Setting | | | | | | | |
| BiasMF | Base | 0.0683 | 0.1039 | **0.0311** | 0.0399 | 0.0449 | 0.0790 | 0.1451 | 0.1375 | 0.0078 | 0.0143 | 0.0046 | 0.0066 |
| | Augment. | **0.0719*** | **0.1353*** | 0.0272 | **0.0411*** | **0.0531*** | **0.0868*** | **0.1761*** | **0.1630*** | **0.0121*** | **0.0198*** | **0.0074*** | **0.0097*** |
| | Improve. | 5.27%↑ | 30.22%↑ | 12.54%↓ | 3.01%↑ | 18.26%↑ | 9.87%↑ | 21.36%↑ | 18.55%↑ | 55.13%↑ | 38.46%↑ | 60.87%↑ | 46.97%↑ |
| NCF | Base | 0.0713 | 0.0985 | **0.0325** | **0.0445** | 0.0581 | 0.0936 | 0.1848 | 0.1721 | 0.0102 | 0.0076 | 0.0188 | 0.0091 |
| | Augment. | **0.0760*** | **0.1233*** | 0.0288 | 0.0414 | **0.0591*** | **0.0968*** | **0.1903*** | **0.1785*** | **0.0133*** | **0.0087*** | **0.0206*** | **0.0108**∗∗ |
| | Improve. | 6.59%↑ | 25.18%↑ | 11.38%↓ | 6.97%↓ | 1.72%↑ | 3.42%↑ | 2.98%↑ | 3.72%↑ | 30.39%↑ | 14.47%↑ | 9.57%↑ | 18.68%↑ |
| LightGCN | Base | 0.0389 | 0.0702 | 0.0150 | 0.0219 | 0.0467 | 0.0815 | 0.1488 | 0.1424 | 0.0096 | 0.0162 | 0.0059 | 0.0076 |
| | Augment. | **0.0788*** | **0.0983*** | **0.0337*** | **0.0384*** | **0.0652*** | **0.1026*** | **0.1703*** | **0.1606*** | **0.0143*** | **0.0225*** | **0.0087*** | **0.0107*** |
| | Improve. | 102.57%↑ | 40.03%↑ | 124.67%↑ | 75.34%↑ | 39.61%↑ | 25.89%↑ | 14.45%↑ | 12.78%↑ | 48.96%↑ | 38.89%↑ | 47.46%↑ | 40.79%↑ |
| SGL | Base | 0.0345 | 0.0708 | 0.0127 | 0.0210 | 0.0277 | 0.0416 | 0.0855 | 0.0762 | 0.0078 | 0.0138 | 0.0050 | 0.0068 |
| | Augment. | **0.0732*** | **0.0967*** | **0.0367*** | **0.0421*** | **0.0788*** | **0.1204*** | **0.1958*** | **0.1831*** | **0.0133*** | **0.0221*** | **0.0080*** | **0.0106*** |
| | Improve. | 112.17%↑ | 36.58%↑ | 188.98%↑ | 100.48%↑ | 184.48%↑ | 189.42%↑ | 129.01%↑ | 140.29%↑ | 70.51%↑ | 60.14%↑ | 60%↑ | 55.88%↑ |
| SimGCL | Base | 0.0421 | 0.0636 | 0.0155 | 0.0212 | 0.0231 | 0.0441 | 0.0810 | 0.0825 | 0.0042 | 0.0078 | 0.0026 | 0.0037 |
| | Augment. | **0.0576*** | **0.0908*** | **0.0232*** | **0.0329*** | **0.0567*** | **0.0908*** | **0.1782*** | **0.1673*** | **0.0128*** | **0.0205*** | **0.0080*** | **0.0099*** |
| | Improve. | 36.82%↑ | 42.77%↑ | 49.68%↑ | 55.19%↑ | 145.45%↑ | 105.90%↑ | 120.00%↑ | 102.79%↑ | 204.76%↑ | 162.82%↑ | 207.69%↑ | 167.57%↑ |
| | | | | | | Zero-Shot Setting | | | | | | | |
| BiasMF | Base | 0.0096 | 0.0165 | 0.0031 | 0.0041 | 0.0311 | 0.0769 | 0.0167 | 0.0292 | 0.0038 | 0.0068 | 0.0020 | 0.0029 |
| | Augment. | **0.0246*** | **0.0373*** | **0.0107*** | **0.0135*** | **0.1381*** | **0.1490*** | **0.0828*** | **0.0584*** | **0.0056*** | **0.0103*** | **0.0026*** | **0.0040*** |
| | Improve. | 156.25%↑ | 126.06%↑ | 245.16%↑ | 229.27%↑ | 344.05%↑ | 93.76%↑ | 395.81%↑ | 100.00%↑ | 47.37%↑ | 51.47%↑ | 30.00%↑ | 37.93%↑ |
| NCF | Base | 0.0301 | 0.0383 | 0.0080 | 0.0097 | 0.0480 | 0.1158 | 0.0196 | 0.0384 | 0.0044 | 0.0022 | 0.0056 | 0.0026 |
| | Augment. | **0.0424*** | **0.0469*** | **0.0112*** | **0.0122*** | **0.1700*** | **0.1774*** | **0.0984*** | **0.0974*** | **0.0051*** | **0.0031*** | **0.0088*** | **0.0041*** |
| | Improve. | 40.86%↑ | 22.45%↑ | 40.00%↑ | 25.77%↑ | 254.17%↑ | 53.20%↑ | 402.04%↑ | 153.65%↑ | 15.91%↑ | 40.91%↑ | 57.14%↑ | 57.69%↑ |
| LightGCN | Base | 0.0138 | 0.0292 | 0.0046 | 0.0078 | 0.0974 | 0.1256 | 0.0446 | 0.0415 | 0.0092 | 0.0160 | 0.0051 | 0.0070 |
| | Augment. | **0.0196*** | **0.0389*** | **0.0064*** | **0.0086*** | **0.1371*** | **0.1453*** | **0.0697*** | **0.0459*** | **0.0133*** | **0.0188*** | **0.0090*** | **0.0106*** |
| | Improve. | 42.03%↑ | 33.22%↑ | 39.13%↑ | 10.26%↑ | 40.76%↑ | 15.68%↑ | 56.28%↑ | 10.60%↑ | 44.57%↑ | 17.50%↑ | 76.47%↑ | 51.43%↑ |
| SGL | Base | 0.0162 | 0.0264 | 0.0062 | 0.0074 | 0.0385 | 0.1441 | 0.0274 | 0.0579 | 0.0065 | 0.0114 | 0.0036 | 0.0050 |
| | Augment. | **0.0254*** | **0.0450*** | **0.0089*** | **0.0107*** | **0.1126*** | **0.1756*** | **0.0384*** | **0.1066*** | **0.0111*** | **0.0176*** | **0.0066*** | **0.0084*** |
| | Improve. | 56.79%↑ | 70.45%↑ | 43.55%↑ | 44.59%↑ | 92.47%↑ | 21.86%↑ | 40.15%↑ | 84.11%↑ | 70.77%↑ | 54.39%↑ | 83.33%↑ | 68.00%↑ |
| SimGCL | Base | 0.0164 | 0.0300 | 0.0055 | 0.0084 | 0.0793 | 0.1259 | 0.0336 | 0.0460 | 0.0078 | **0.0140** | 0.0042 | 0.0059 |
| | Augment. | **0.0312*** | **0.0388*** | **0.0098*** | **0.0115*** | **0.1508*** | **0.1895*** | **0.1550*** | **0.1647*** | **0.0084*** | 0.0137 | **0.0044*** | 0.0059 |
| | Improve. | 90.24%↑ | 29.33%↑ | 78.18%↑ | 36.90%↑ | 90.16%↑ | 50.52%↑ | 361.31%↑ | 258.04%↑ | 7.69%↑ | 2.14%↓ | 4.76%↑ | —— |

---

[1] https://msnews.github.io

[2] https://www.kaggle.com/datasets/netflix-inc/netflix-prize-data

To demonstrate the effectiveness of our RecLM in enhancing performance, particularly in cold-start scenarios, we apply it to five common collaborative filtering methods. The "full-shot" setting corresponds to the complete dataset, while the "zero-shot" setting refers to the pure cold-start condition. The ***Base*** variant applies the cold-start recommendation paradigm to the baseline recommenders without any profiling enhancement via LLMs, whereas the ***Augment*** variant integrates RecLM into the base recommenders. Detailed settings and implementation information are provided in Appendices A.3 and A.5. The evaluation results in Table 1 reveal several interesting observations.

**(i) Performance Improvement in Integrated Recommenders**. We consistently find that integrating RecLM with backbone recommenders leads to enhanced performance compared to the base variant, which relies on raw external item features and ID-based user embeddings in both **supervised** and **zero-shot** settings. This provides compelling evidence for the effectiveness of RecLM. We attribute these improvements to two key factors: *First*, for supervised recommendation scenarios, RecLM leverages instruction-tuned LLMs to generate accurate user and item profiles as auxiliary information, effectively enhancing the semantic representation of user preference. *Second*, our tuning paradigm guides the LLMs in capturing user collaborative relationships, allowing for the generation of high-quality, personalized profiles that demonstrate strong generalization in zero-shot scenarios.

**(ii) Outstanding Performance in Cold-Start Scenarios.** This improvement arises from our innovative modifications to the ID-embedding paradigm employed in current recommenders. By incorporating external features specifically designed to address the challenges of interaction data scarcity, we have significantly enhanced the effectiveness of these systems. Remarkably, we observe substantial performance improvements even in the relatively sparser MIND and Industrial datasets, where data limitations traditionally pose significant hurdles. By leveraging our RecLM for user and item profiling, we significantly enhance the generalization capabilities of existing recommenders.

**(iii) Practicality and Scalability for Real-World Deployment.** The results from the Industrial dataset demonstrate that RecLM consistently enhances the performance of recommenders in large-scale, highly sparse real-world scenarios. Furthermore, our user and item profile generation methods can be efficiently executed as an offline profiling system to support online applications, making them highly practical for real-world recommendations. To facilitate online recommendation systems, user and item profiles can be updated at regular intervals, such as daily or weekly. The performance improvements observed across various backbone models indicate that RecLM can easily adapt to a range of business models, significantly enhancing their overall effectiveness.

## 4.3 ABLATION STUDY (RQ2)

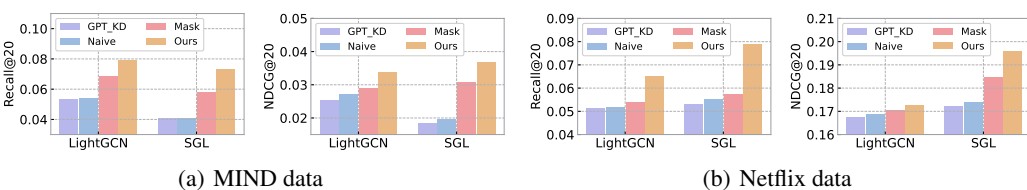

(a) MIND data                                    (b) Netflix data

Figure 2: Ablation study on the LLM tuning techniques in the RecLM framework.

We conducted extensive experiments to validate the effectiveness of our proposed instruction tuning techniques by customizing three variants of RecLM: *GPT_KD*, *Naive*, and *Mask*. Detailed descriptions of these variants can be found in Appendix A.4. The results of our experiments are illustrated in Figure 2, allowing us to draw the following conclusions:

**(i) Advantage of Collaborative Instruction Tuning.** The results in Figure 2 show that using instruction tuning to capture collaborative relationships among users and items, along with the masking tuning strategy (*Mask*), significantly enhances performance compared to *GPT_KD*. This improvement suggests that our tuning solution generates more precise, high-quality profiles by leveraging collaborative information effectively. In contrast, profiling based solely on user interaction history has limitations, as it lacks the guidance from collaborative insights. Consequently, this approach often results in less accurate profiles that may include noisy interaction records.

**(ii) Effectiveness of the Masking-Based Tuning Strategy.** Although the *Naive* variant also employs a two-round dialogue-based instruction tuning technique similar to the *Mask* variant, its improvement over the *GPT_KD* variant is limited. This underscores the advantages of the masking-based tuning strategy, which effectively utilizes responses from the two-round dialogue to update the weights of the LLM and guide its learning of collaborative relationships between users.

**(iii) Benefits of Reinforcement Learning-Based Feature Enhancement.** The results indicate that the *Mask* variant performs significantly worse than RecLM. This finding suggests that the proposed reinforcement learning (RL)-based personalized feature enhancement technique effectively addresses the noise issues and over-smoothing problems associated with the collaborative instruction-tuning paradigm. As a result, it enables the LLM to generate more accurate and personalized profiles.

### 4.4 EFFECTIVENESS OF LLM-EMPOWERED PROFILING SYSTEM IN RECLM (RQ3)

To investigate the impact of our LLM-empowered profiling system on user and item feature enhancements, we developed two variants of RecLM: one that excludes user feature enhancement (denoted as *i.e., w/o User Aug.*) and another that excludes item feature enhancement (denoted as *i.e., w/o Item Aug.*). The experiments were conducted on the MIND and Netflix datasets using the full-shot setting, with LightGCN and SGL as the backbone models. The evaluation results are presented in Table 2, allowing us to draw the following conclusions.

Table 2: Performance *w.r.t.* various aug. variants.

| Dataset | | MIND | | Netflix | |
|---|---|---|---|---|---|
| Backbone | Variants | R@20 | N@20 | R@20 | N@20 |
| LightGCN | Base | 0.0389 | 0.0150 | 0.0467 | 0.1488 |
| | w/o User Aug. | 0.0302 | 0.0123 | 0.0384 | 0.1213 |
| | w/o Item Aug. | 0.0719 | 0.0287 | 0.0505 | 0.1621 |
| | RecLM | **0.0788** | **0.0337** | **0.0652** | **0.1703** |
| SGL | Base | 0.0345 | 0.0127 | 0.0277 | 0.0855 |
| | w/o User Aug. | 0.0253 | 0.0093 | 0.0173 | 0.0578 |
| | w/o Item Aug. | 0.0719 | 0.0289 | 0.0502 | 0.1546 |
| | RecLM | **0.0732** | **0.0367** | **0.0788** | **0.1958** |

**(i) User-Side Feature Enhancement.** The exclusion of user-side feature enhancements (denoted as *i.e., w/o User Aug.*) results in a significant decline in performance across both evaluated datasets and backbone models. This underscores the critical role of our RecLM as the profiling system for improving performance. Relying solely on the original ID embedding for the user side is insufficient for effectively capturing and modeling user preferences. We attribute this outcome to both the effective extraction of text features and the successful integration of graph and textual information.

**(ii) Item-Side Feature Enhancement.** The exclusion of item-side feature enhancements (denoted as *i.e., w/o Item Aug.*) also leads to a noticeable decline in the recommender's performance. Interestingly, when item-side feature enhancements are retained without incorporating any user-side feature enhancements (denoted as *i.e., w/o User Aug.*), the performance can drop even below that of the Base variant. This discrepancy can be attributed to the interplay between raw and enhanced features on the item side, which creates a complex dynamic. Relying solely on ID embedding for the user side proves inadequate for effectively modeling user preferences.

### 4.5 TRAINING EFFICIENCY ANALYSIS OF RECLM (RQ4)

To evaluate the efficiency of our RecLM approach, we conduct both a theoretical complexity analysis and an empirical running time test. **Theoretical Analysis**: The time complexity of the MLP used to transfer textual features $\mathbf{f} \in \mathbb{R}^{d_t}$ of items into the model's latent space $\mathbb{R}^d$ is $\mathcal{O}(N \times (d_t \times d + d \times d))$, where $N$ represents the number of nodes, and $d_t$ and $d$ denote the dimensionalities of the original text features and the latent space, respectively. **Empirical Evaluation**: We present the per-epoch training time in Table 3. The evaluation was conducted on a server equipped with NVIDIA A100 GPUs (40 GB memory). The results indicate that for larger models (e.g., GNN-based methods), our RecLM requires relatively little

Table 3: Training efficiency *w.r.t.* integration with various recommenders.

| Dataset | Recommender | Base | RecLM | Cost |
|---|---|---|---|---|
| MIND | BiasMF | 0.72s | 0.85s | +18.06% |
| | NCF | 0.76s | 0.85s | +11.84% |
| | LightGCN | 0.79s | 0.86s | +8.86% |
| | SGL | 1.93s | 2.01s | +4.15% |
| | SimGCL | 2.63s | 2.69s | +2.28% |
| Netflix | BiasMF | 14.38s | 16.42s | +14.19% |
| | NCF | 15.02s | 17.17s | +14.31% |
| | LightGCN | 20.47s | 20.95s | +2.34% |
| | SGL | 64.98s | 65.08s | +0.15% |
| | SimGCL | 44.02s | 44.61s | +1.34% |
| Industrial | BiasMF | 7.07s | 8.85s | +25.18% |
| | NCF | 7.58s | 8.45s | +11.48% |
| | LightGCN | 9.33s | 10.25s | +9.86% |
| | SGL | 32.34s | 32.87s | +1.64% |
| | SimGCL | 85.41s | 86.52s | +1.30% |

additional time, often falling below 10%. In denser datasets like Netflix, this additional time can be reduced to under 5%. Even for smaller recommenders, the maximum additional time is approximately 25%. Given the substantial improvements in recommendation performance provided by our method, the incurred costs are considered acceptable.

### 4.6 COMPARISON WITH EXISTING LLM-ENHANCED METHODS(RQ5)

We further compare RecLM with the existing work LLMRec Wei et al. (b), which also enhances recommendation systems using LLMs, to highlight the superiority of our proposed instruction-tuning technique. The experimental results are presented in Table 4. Specifically, LLMRec generates profiles for items and users by directly calling the LLM's API without fine-

Table 4: Performance Comparison with LLMRec.

| Dataset | | MIND | | Netflix | |
|---|---|---|---|---|---|
| Backbone | Variants | R@20 | N@20 | R@20 | N@20 |
| LightGCN | Base | 0.0389 | 0.0150 | 0.0467 | 0.1488 |
| | w/ LLMRec | 0.0532 | 0.0254 | 0.0515 | 0.1674 |
| | w/ RecLM | **0.0788** | **0.0337** | **0.0652** | **0.1703** |
| SGL | Base | 0.0345 | 0.0127 | 0.0277 | 0.0855 |
| | w/ LLMRec | 0.0405 | 0.0185 | 0.0529 | 0.1721 |
| | w/ RecLM | **0.0732** | **0.0367** | **0.0788** | **0.1958** |

tuning for the profile generation task. This approach fails to effectively leverage the collaborative relationships among users. As a result, RecLM demonstrates significant performance advantages across two public datasets, leading to notable improvements in the performance of the base models.

### 4.7 CASE STUDY(RQ6)

To intuitively explore the contribution of reinforcement learning to the personalization of generated profiles, we conducted a case study using the MIND dataset. In this study, as shown in Figure 3, the target user for whom the profile is being generated is *User 49*. This user has interacted with two items: *Item 472* and *Item 1572*. Additionally, we identified three similar users who provide collaborative information: *User 11451*, *User 20522*, and *User 341*.

The user profile generated for *User 49* after instruction tuning, but without reinforcement learning (RL) tuning, contains several irrelevant keywords related to the interacted items, such as "Foodie," "Food and drink," and "Horoscope." Notably, these

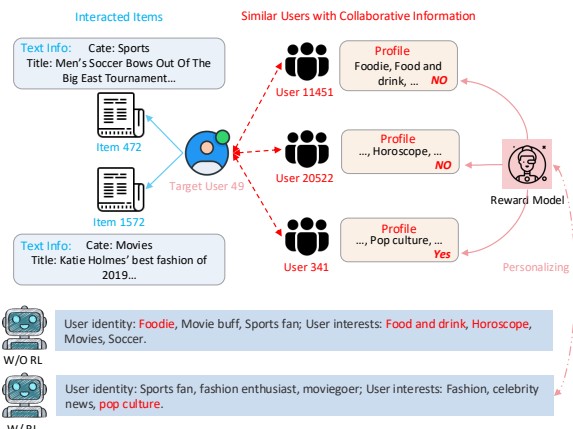

Figure 3: Generated profiles w/ and w/o RL.

terms also appear in the profiles of *User 11451* and *User 20522*, suggesting that the generated profile is overly influenced by too many collaborative users. In contrast, the profile generated for *User 49* after RL tuning effectively preserves the preferences indicated in the interaction history while incorporating relevant implicit keywords from collaborative users. For example, the term "pop culture" is derived from *User 341*'s profile. This approach provides precise and valuable additional information for modeling *User 49*'s preferences. We attribute this improvement to our proposed RL-based personalized feature enhancement techniques, which effectively address the noise and over-smoothing issues that can arise during the instruction-tuning process.

## 5 CONCLUSION

In this work, we introduce RecLM, a groundbreaking model-agnostic recommendation instruction-tuning paradigm that seamlessly integrates large language models (LLMs) with collaborative filtering techniques to significantly enhance user profiling, especially in cold-start scenarios. This innovative approach leverages LLMs to generate rich user and item profiles by harnessing collaborative relationships and textual features, effectively tackling the critical challenges of data sparsity and noise. Furthermore, we incorporate a unique reinforcement learning mechanism to refine profile quality and optimize recommendation outputs, enabling substantial performance gains across diverse recommender systems. This combination of techniques not only enhances the robustness of the recommendations but also ensures scalability and adaptability in real-world applications.

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

# A  APPENDIX / SUPPLEMENTAL MATERIAL

## A.1  DETAILS OF DATASET

Table 5 provides a summary of the statistical information for the three datasets. The following sections outline the specific details for each dataset:

- **MIND**: This large-scale dataset is designed for news recommendation research. We selected data from two consecutive days, assigning one day as the training set and the other as the test set. The raw text includes the news category, title, and abstract.

- **Netflix**: It is selected from a renowned video streaming platform, and we get the implicit feedback data from the Netflix Prize Data on Kaggle. We curated two consecutive years' worth of data based on time, utilizing one year as the training set and the other as the test set. The raw text information for the items was derived from the movie titles themselves.

- **Industrial**: It is a large-scale real dataset, which is collected from a prominent online content platform (name omitted for anonymity), serving millions of users. It comprises news articles. We sampled data from two consecutive dates, assigning them as the training set and test set, respectively. The raw text information for each item is represented by its title.

Table 5: Statistics of the experimental datasets.

| *Statistics* | **MIND** | **Netflix** | **Industrial** |
|---|---|---|---|
| **# User** | 57128 | 16835 | 117433 |
| **# Overlap. Item** | 1020 | 6232 | 72417 |
| **# Snapshot** | daily | yearly | daily |
| Training Set | | | |
| **# Item** | 2386 | 6532 | 152069 |
| **# Interactions** | 89734 | 1655395 | 858087 |
| **# Sparsity** | 99.934% | 98.495% | 99.995% |
| Test Set | | | |
| **# Item** | 2461 | 8413 | 158155 |
| **# Interactions** | 87974 | 1307051 | 876415 |
| **# Sparsity** | 99.937% | 99.077% | 99.995% |

## A.2  DETAILS OF SELECTED BASE MODELS

This section gives a brief introduction of the selected base models in this work.

- **BiasMF** Koren et al. (2009): It is a matrix factorization method that aims to enhance user-specific preferences for recommendation by incorporating bias vectors for users and items.

- **NCF** He et al. (2017): It is a neural network-based method that replaces the dot-product operation in conventional matrix factorization with multi-layer neural networks. This allows the model to capture complex user-item interactions and provide recommendations. For our comparison, we utilize the NeuMF variant of NCF.

- **LightGCN** He et al. (2020): This model leverages the power of neighborhood information in the user-item interaction graph by using a layer-wise propagation scheme that involves only linear transformations and element-wise additions.

- **SGL** Wu et al. (2021): The model enhances LightGCN by integrating contrastive learning with self-supervision. It employs data augmentation strategies, including random walks and node/edge dropout, to corrupt graph structures.

- **SimGCL** Yu et al. (2022): This work introduces a straightforward contrastive learning (CL) method that eliminates graph augmentations. Instead, it adds uniform noise to the embedding space to generate contrastive views.

## A.3  PERFORMANCE COMPARISON: SETTING

In the performance comparison experiments outlined in Sec. 4.2, we considered two distinct testing data settings: the full-shot setting and the zero-shot setting. The full-shot setting entailed using the

original test set as the testing data, where certain items in the test set had appeared in the training set previously. Conversely, the zero-shot setting involved exclusively testing items that had not been encountered in the training set. This setting was specifically designed to assess the effectiveness of our proposed RecLM in addressing the item cold-start scenario, where limited or no prior information is available for certain items.

In the conducted experiments, we explored two variants: *Base* and *Augment*. The *Base* variant demonstrates the application of our proposed cold-start recommendation paradigm by utilizing only user-side ID embeddings and item-side raw text embeddings, without incorporating the profiles generated by LLMs. On the other hand, the *Augment* variant involves fully integrating our proposed RecLM into traditional recommenders. The comparison between two variants enables us to assess the effectiveness of our approach in enhancing the performance of recommenders by leveraging LLMs to generate informative profiles.

### A.4 ABLATION STUDY: SETTING

In the case of *GPT_KD* variant, the approach involves exclusively fine-tuning the open-source LLM by utilizing user profile data generated solely through ChatGPT3.5, as discussed in Sec. 3.3.1. Conversely, for *Naive* variant, the two-turn dialogue-based instruction tuning technique (*i.e.,* Sec. 3.3.2) is applied based on the variant *GPT_KD*, but with the tuning strategy limited to the conventional single-turn dialogue tuning approach. As for the variant *Mask*, a similar two-turn dialogue-based instruction tuning technique is employed based on the variant *GPT_KD*, with the additional application of a masking-based tuning strategy. As for *Ours*, it refers to RecLM, which employs RL-based personalized feature enhancement based on the variant *Mask*.

### A.5 IMPLEMENTATION DETAILS

#### A.5.1 PARAMETER-EFFICIENT FINE-TUNING

To achieve efficient fine-tuning of LLMs while preserving their inherent knowledge reasoning capabilities, we employed the Parameter-Efficient Fine-Tuning (PEFT) method. Specifically, in this study, we chose Low-Rank Adaptation (LoRA) Hu et al. (2021) as the fine-tuning technique for the open-source LLMs, specifically Llama2-7b-chat Touvron et al. (2023). This approach allows us to strike a balance between retaining the valuable knowledge of the pre-trained models and adapting them to specific tasks effectively.

#### A.5.2 INTEGRATION OF RECLM INTO VARIOUS BASE RECOMMENDERS

Following the integration of our method into various base recommenders, we meticulously conducted an extensive hyperparameter search, and also explored the optimal approach for incorporating profile features for each recommendation methods, ensuring a fair comparison. Specifically, each base model is implemented with PyTorch, using Adam optimizer and Xavier initializer with default parameters. Training batch size is set as 4096. The dimensionality of embedding vectors is set as 32. The learning rate is set as $1e-3$. The coefficient for controling $\mathcal{L}_2$ regularization term is searched in $\{1e-3, 1e-4, 1e-5, 1e-6, 1e-7\}$. For GNN-based models (*e.g.,* LightGCN, SGL, and SimGCL), the number of GCN layers is set as 2. For SSL-based models (*e.g.,* SGL and SimGCL), the temperature coefficient is searched in $\{0.1, 0.5, 1.0\}$.

### A.6 INSTRUCTION DESIGNS

In this section, we provide a comprehensive overview of the instructions utilized for fine-tuning at each stage of our process. We will also discuss the methodologies employed to construct both positive and negative training samples for the reinforcement learning reward model.

- **Instruction designs for ChatGPT knowledge distillation.** As shown in Figure 4, to facilitate the knowledge distillation process of ChatGPT, we leverage the textual information associated with each user and the items they interact with as inputs for the LLMs. The LLMs then generate user profiles, encompassing the user's identity along with their respective interests.

Now you are a user profile generator. I will provide you with a list of news articles that a user has clicked on in the past. Each news article contains four pieces of information: category, subcategory, title, and abstract. Based on this information, please generate the user's profile. Here is the list of previously clicked news articles:: [Item Text Info.1], [Item Text Info.2], ..., [Item Text Info.N]. Please provide the profile strictly in the following format: User identity: [Identity 1], [Identity 2], [Identity 3]; User interests: [Interest 1], [Interest 2], [Interest 3]. Emphasize that only the most likely three identities and interests should be provided, and strictly adhere to the above format.

User identity: [Identity 1], [Identity 2], [Identity 3]; User interests: [Interest 1], [Interest 2], [Interest 3].

**Instruction Designs for GPT_KD**

Figure 4: Instruction designs for ChatGPT knowledge distillation.

- **Instruction designs for two-turn dialogue instruction tuning.** For the instruction-tuning based on two-round dialogues, meticulous attention has been given to designing corresponding instructions. As illustrated in Figure 5, we commence by providing specific system instructions to stimulate the LLMs' comprehension of collaborative filtering methods. Subsequently, in the first round of dialogue, the input instructions encompass the interaction history of several similar users, along with the relevant details of the items involved. To obtain those similar users, we employ a conventional ID-based collaborative filtering recommendation system, followed by similarity calculation based on these embeddings. The expected output from the LLMs should include user profiles for each mentioned user in the input. Moving on to the second round of dialogue, we explicitly prompt the LLMs to determine, based on the acquired user profiles and item information, whether a previously mentioned item is likely to be interacted with by a specific user using collaborative filtering methods. The expected response from the LLMs should be a binary "Yes" or "No" answer.

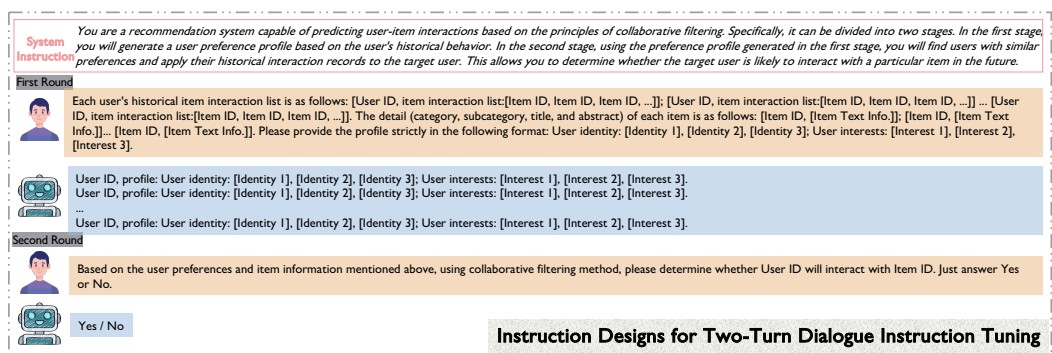

Figure 5: Instruction designs for two-turn dialogue instruction tuning.

- **Instruction designs for user profile generation.** Once the instruction-tuning stage is complete, the LLMs are equipped with the capability to generate profiles while considering collaborative relationships. In line with Figure 6, we have meticulously designed instructions specifically for user profile generation. Consistent with the instruction-tuning stage, we provide explicit system instructions to stimulate the LLMs' comprehension of collaborative filtering methods. The input instructions encompass the interaction records of multiple similar users (including a target user for whom the LLMs are required to generate a profile) as well as detailed textual information pertaining to the involved items. The expected output from the LLMs is the target user profile.

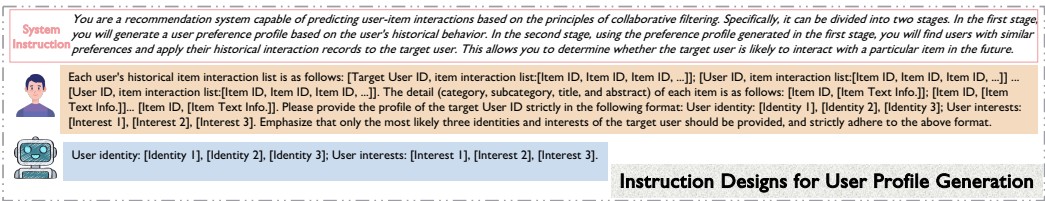

Figure 6: Instruction designs for user profile generation.

- **Instruction designs for item profile generation.** To ensure semantic alignment between user-side and item-side features, our next objective, after obtaining high-quality user profiles, is to generate

item profiles based on the user's profile. Here, the item profile refers to the profile of the target user for that particular item. To accomplish this, we adopt a two-step approach. Firstly, for items that have user interactions, we generate item profiles by leveraging the profiles of the interacting users. This helps establish a connection between the users and the items they engage with. Secondly, using the raw embeddings of the items, we search for similar cold-start items and employ the LLM to infer their profiles based on semantic similarity. As depicted in Figure 7, the input instructions consist of a target item and several similar items. We provide the specific textual information of these items, along with the profiles of the similar items (selected from items that already have profiles). The expected output from the LLMs is the profile of the target item, further enhancing semantic alignment across the recommendation system.

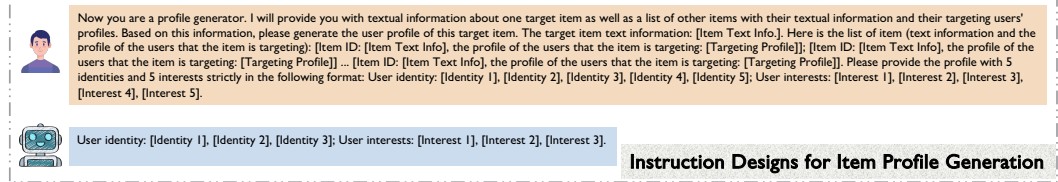

Figure 7: Instruction designs for item profile generation.

- **Positive/Negative responses construction for reward model training.** In Sec 3.4, we propose personalized feature enhancement based on reinforcement learning as a means to address the noise introduced by instruction-tuning and the potential over-smoothing issue stemming from collaborative feature enhancement. The crux of reinforcement learning lies in training the reward model, and constructing high-quality positive and negative samples plays a pivotal role in this process. As shown in Figure 8, for positive samples, we leverage SOTA LLMs (*e.g.*, ChatGPT) with a manual selection approach. For negative samples, they can be categorized into two distinct groups. The first category consists of profiles of similar users, which aim to train the reward model in distinguishing more nuanced profiles and mitigating the over-smoothing issue. The second category encompasses low-quality responses of various types, such as missing or repeated profiles, thereby providing negative examples for training the reward model effectively.

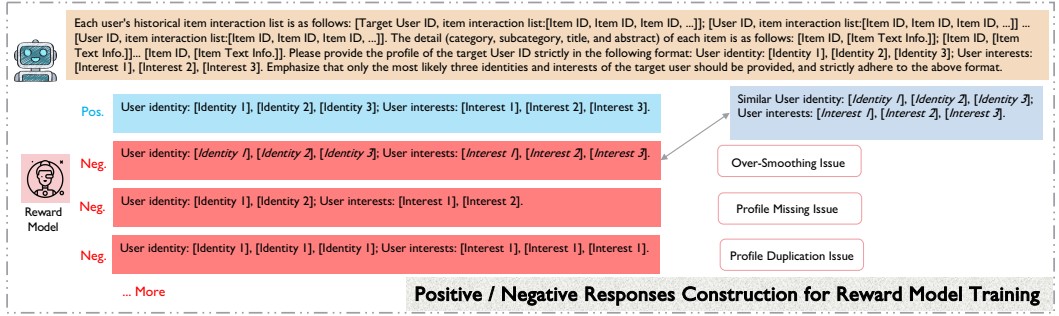

Figure 8: Positive/Negative responses construction for reward model training.

## A.7 LIMITATIONS AND BROADER IMPACTS

In real-world scenarios, items commonly have abundant modal information, including text, images, audio, and more. However, this work primarily focuses on exploring the collaborative feature enhancement paradigm based on textual features, and does not fully exploit the potential of multi-modal information. While the proposed method can be extended to other modalities using distinct modal encoders, it is important to note that other modalities may introduce novel challenges and opportunities for feature enhancement. Thus, the exploration of these modalities represents a promising future direction for further investigation.

