# OpenReview forum: "RecLM: Recommendation Instruction Tuning with Large Language Models"
_ICLR.cc/2025/Conference — ICLR 2025 Conference Withdrawn Submission_

### Official Review · Reviewer_RCLa · 2024-10-31

**Soundness:** 3
**Presentation:** 3
**Contribution:** 2
**Rating:** 5
**Confidence:** 4

**Summary:**

This paper presented RecLM, a model-agnostic framework that integrated large language models (LLMs) with collaborative filtering to improve recommendation accuracy in sparse-data and cold-start scenarios by capturing diverse user preferences across multiple attributes. Using instruction tuning with two-turn dialogue and reinforcement learning, RecLM generated personalized, high-quality profiles that enhanced the performance of existing recommendation systems. Experimental results showed that RecLM significantly improved accuracy and fairness across various datasets, demonstrating its adaptability and effectiveness in real-world applications.

**Strengths:**

1. The paper introduced RecLM, a model-agnostic framework that improved user profiling in recommendation systems, particularly in cold-start scenarios, by integrating large language models (LLMs) with collaborative filtering.

2. It presented a two-turn dialogue-based instruction tuning method, leveraging collaborative relationships to refine profiles through both interaction data and similarity-based signals, enhancing profile accuracy and recommendation quality.

3. Extensive evaluations across various datasets, including industrial applications, demonstrated RecLM’s effectiveness in improving recommendation performance in both supervised and zero-shot scenarios.

4. The authors used reinforcement learning to further enhance personalization and reduce noise, ensuring RecLM’s scalability and practical applicability in large-scale settings.

**Weaknesses:**

1. The current method generates user and item profiles mainly based on textual features and does not consider multimodal information, while the LLMRec compared in RQ5 considers multimodal information.

2. Although the paper tests on various publicly available datasets, the selected datasets (e.g., Netflix, MIND) are primarily text-rich, thus limiting the demonstration of performance on datasets that are text-sparse or dominated by multimodal information. Furthermore, while the comparison methods include several typical collaborative filtering and graph neural network models, they lack a broader selection of diverse recommendation algorithms.

3. For the training of reward models in reinforcement learning methods relies on high-quality positive and negative samples, where the positive samples are generated via ChatGPT, while the negative samples include profiles of similar users and multiple types of low-quality responses. If these negative samples do not effectively represent the low-quality outputs that the model may generate in real-world applications, does the reward model still have some validity.

**Questions:**

1. Given the extensive research on LLM-based enhancement, why does the comparison in RQ5 only focus on existing LLM-enhanced methods? And why don’t the main experiments include a wider range of benchmark models to fully demonstrate the validity of the proposed methods?

2. In this paper, reinforcement learning is utilized in the denoising stage, what are the advantages of reinforcement learning over other denoising methods applied to the proposed method in this paper? The introduction of reinforcement learning increases the model complexity, and if the reward model is not well designed, is it likely to lead to biased profile generation.

---

### Official Review · Reviewer_w1fs · 2024-11-02

**Soundness:** 1
**Presentation:** 1
**Contribution:** 2
**Rating:** 3
**Confidence:** 4

**Summary:**

This paper introduces a model-agnostic recommendation instruction-tuning paradigm, RecLM, which finetunes large language models to summarize user profiles for cold start collaborative filtering. RecLM is tested as a plug-in component for various collaborative filtering backbones, and experiments demonstrate its effectiveness.

**Strengths:**

1. The motivation of this work, enhancing user's profile for cold-start scenario, is interesting.
2. The method is easy to understand.

**Weaknesses:**

1.	The literature review in this paper is extremely insufficient, raising questions about the contribution of this work.
On the one hand, there are only two references in the introduction, which is extremely unprofessional for an academic paper. Please refer to literatures for how to write the introduction in an academic manner correctly [1, 2].
On the other hand, while the main claims of this paper focus on the cold-start scenario, this paper lacks discussion with important and relevant reference papers [3, 4, 5, 6, 7] in this area. Moreover, progresses of adapting LLMs for cold-start recommendation are also totally ignored in this paper [8, 9].
The lack of literature review results raises the concern about the incorrect understanding of cold-start recommendation of this paper, therefore reducing the novelty and contribution of this paper.
2.	The experiments in this paper fail to support the authors' claims. The authors state, “We propose the development of effective language models as profiling systems specifically designed for recommendation tasks, aimed at enhancing performance in cold-start recommendation scenarios.” However, this paper does not include any comparison or discussion with other important cold-start recommenders (e.g., Dropoutnet [6] and CCFCRec [7]).
3.	Baselines for user profile modeling are missing. For instance, RLMRec [10] also models user profiles, while this paper does not compare with it as a baseline.
4.	The experimental setup and results are questionable. The metrics for the baseline selected in this paper (i.e., LLMRec [11]) show a significant decline compared to those reported in the original paper. In LLMRec [11], the reported results on the Netflix dataset are Recall@20=0.0829 and NDCG@20=0.0347. However, in the reproduction of LLMRec results in this paper, Recall@20 is 0.0529 and NDCG@20 is 0.1721, indicating a significantly lower Recall result and an unusually high NDCG result. Additionally, it is noteworthy that the Recall@20 reported in the original paper of LLMRec, 0.0829, is higher than that of the RecLM method proposed by the authors, which achieved Recall@20=0.0788. Please specify the experimental setting difference between this paper and LLMRec.

5.	The writing of this paper can be significantly improved.

    $\bullet$  The citation format in this paper is incorrect, making this paper extremely unprofessional. For citations that are not used as nouns, please use \citep instead of \cite for referencing. For example, in line 94, the correct citation format should be “methods, including NGCF (Wang et al, 2019), GCCF (Chen et al., 2020), and LightGCN (He et al., 2020).” For more details, please refer to the APA citation format on Wikipedia.

   $\bullet$The introduction is poorly written, with little effort to summarize the current work before directly transitioning to the contributions. Moreover, important references are missing, as I have emphasized in the weakness 1.

  $\bullet$The writing about the implementation details of RecLM is unclear. This paper aims to design a plug-in module, but the methodology section only explains how to fine-tune the module while failing to discuss how the module can be plugged into an existing recommendation model. Important implementation details such as the loss function for integration should be incorporated.


[1] He, Xiangnan, et al. "Lightgcn: Simplifying and powering graph convolution network for recommendation." Proceedings of the 43rd International ACM SIGIR conference on research and development in Information Retrieval. 2020.

[2] Kang, Wang-Cheng, and Julian McAuley. "Self-attentive sequential recommendation." 2018 IEEE international conference on data mining (ICDM). IEEE, 2018.

[3] Schein, Andrew I., et al. "Methods and metrics for cold-start recommendations." Proceedings of the 25th annual international ACM SIGIR conference on Research and development in information retrieval. 2002.

[4] Li, Jingjing, et al. "From zero-shot learning to cold-start recommendation." Proceedings of the AAAI conference on artificial intelligence. Vol. 33. No. 01. 2019.

[5] Wei, Yinwei, et al. "Contrastive learning for cold-start recommendation." Proceedings of the 29th ACM International Conference on Multimedia. 2021.

[6] Volkovs, Maksims, Guangwei Yu, and Tomi Poutanen. "Dropoutnet: Addressing cold start in recommender systems." Advances in neural information processing systems 30 (2017)

[7] Zhou, Zhihui, Lilin Zhang, and Ning Yang. "Contrastive collaborative filtering for cold-start item recommendation." Proceedings of the ACM Web Conference 2023. 2023.

[8] Sanner, Scott, et al. "Large language models are competitive near cold-start recommenders for language-and item-based preferences." Proceedings of the 17th ACM conference on recommender systems. 2023.

[9] Huang, Feiran, et al. "Large Language Model Interaction Simulator for Cold-Start Item Recommendation." arXiv preprint arXiv:2402.09176 (2024).

[10] Ren, Xubin, et al. "Representation learning with large language models for recommendation." Proceedings of the ACM on Web Conference 2024. 2024.

[11] "Llmrec: Large language models with graph augmentation for recommendation." Proceedings of the 17th ACM International Conference on Web Search and Data Mining. 2024.

**Questions:**

See in Weaknesses

---

### Official Review · Reviewer_WWgm · 2024-11-05

**Soundness:** 3
**Presentation:** 3
**Contribution:** 2
**Rating:** 6
**Confidence:** 4

**Summary:**

The paper introduces RecLM, a model-agnostic recommendation instruction-tuning framework that leverages Large Language Models (LLMs) to enhance existing recommender systems. The key idea is using LLMs to generate high-quality user and item profiles by capturing collaborative relationships and integrating them with traditional collaborative filtering approaches. The framework includes a two-turn dialogue-based instruction tuning paradigm and reinforcement learning-based feature enhancement to improve profile quality and personalization.

**Strengths:**

- Present an interesting approach combining LLMs with collaborative filtering in a model-agnostic way.
- Comprehensive experiments across multiple datasets demonstrate the effectiveness of the proposed method.
- Model-agnostic design that can enhance existing recommender systems.

**Weaknesses:**

- Could include more comparisons with other LLM-based recommendation approaches. I acknowledge that most existing methods are not "model-agnostic", but it would be interesting to see whether the proposed method can achieve comparable results with other methods that utilize collaborative and semantic signals.
- The proposed method still may not effectively handle cold-start users, since it relies on user profiles.
- It would be better if the authors discuss more on the computational efficiency, since LLMs are much more expensive than traditional recommender systems. Constructing good SFT and PPO data and updating them timely may also be difficult for practical recommender systems.

**Questions:**

- How does the system handle dynamic user preferences that change over time? Is there a mechanism for updating profiles efficiently?
- What is the minimum amount of interaction data needed for the system to finetune to generate meaningful profiles?
- How does the quality of the base LLM affect the final recommendation performance? What are the minimum requirements for the LLM (I see you use the chat model, how about using the base model)? Does the quality of the teacher LLM (i.e., GPT-3.5, which is a weak model in late 2024) affect the final performance?
- How does the system handle conflicting or inconsistent user behaviors when generating profiles?

---

### Official Review · Reviewer_8dat · 2024-11-08

**Soundness:** 2
**Presentation:** 1
**Contribution:** 2
**Rating:** 3
**Confidence:** 5

**Summary:**

This paper focuses on leveraging the knowledge from LLM to enhance the traditional collaborative filtering (CF) methods in recommender systems. It proposes a two-step profile generation method called RecLLM based on the instruction-tuning paradigm. The core idea is to leverage collaborative signals to enrich user and item profiles, employing a reinforcement learning-based reward function to further refine these profiles. The motivation behind using information from similar users or items in profile generation is to address the cold-start problem, while reinforcement learning-based personalized feature enhancement aims to improve the quality of profiles derived from noisy data.

**Strengths:**

S1: **Scope**.
This paper focuses on a crucial problem as well as a promising research direction in the recommendation field: enhancing traditional domain-specific CF recommenders by integrating LLM knowledge, i.e., LLM-enhanced recommendation.

S2: **Idea**. The proposed RecLLM for generating a useful and high-quality profile is reasonable. It Aggregates the information from high-order propagation by leveraging the id-based representations as well as utilizing the reward function to fine-tune LLM for better profile generation.

S3: **Experiments**. Results from five CF backbone methods are provided to verify the effectiveness of RecLM. Additionally, the paper includes testing on an unknown industrial dataset (maybe online testing?).

**Weaknesses:**

W1: **Scope of claim**. The proposed method is an LLM-enhanced recommendation method, which utilizes LLM to augment external features to enhance the CF methods. However, the title "RecLM: Recommendation Instruction Tuning with Large Language Models" and the model name "RecLM" seem to claim an LLM-based recommendation method. This discrepancy appears throughout the paper, particularly in the introduction, related work, and methodology sections.

W2: **Experiments**. The baseline selection is far more sufficient as only one LLM-enhanced method (LLMRec) is compared.

W3: **Presentation quality**. The presentation of the paper needs to be improved, especially in sections covering motivation, mathematical formulation, and methodological descriptions.

W4: **Literature reviews**. The literature review is incomplete, with several relevant important references missing.

**Questions:**

Overall, this paper focuses on an important problem and proposes a promising research direction by utilizing a reinforcement learning pipeline to fine-tune the user profiles to enrich the traditional CF methods. This is a promising perspective and worth further investigation. However, I think the current version has clear weaknesses regarding the scope of claims, experimental validation, presentation quality, and literature review completeness.

**Regarding the scope of claims**

Q1: The title, "Recommendation Instruction Tuning with Large Language Models," implies a fully LLM-based recommendation method, similar to approaches like TALLRec. To better reflect the actual method, it would be more accurate to include terms like profile generation and collaborative filtering in the title.

Q2: The primary motivation for leveraging collaborative signals is to address the cold-start problem, a limitation of traditional CF methods. However, as collaborative signals are extracted from conventional CF methods, this creates a contradiction that should be clarified. Consider revising the motivation to accurately reflect the method's design and limitations.

**Regarding the experimental validation and literature review completeness**

Q3: In Section 2 (Related Work), RecLLM is positioned as an LLM-enhanced recommendation method, but the literature review is insufficiently comprehensive. Several relevant works in LLM-enhanced recommendation, such as KAR [1], MoRec [2], AlphaRec [3], and TCF [4], should be included.

[1] Towards Open-World Recommendation with Knowledge Augmentation from Large Language Models.

[2] Where to Go Next for Recommender Systems? ID- vs. Modality-based Recommender Models Revisited.

[3] Language Representations Can Be What Recommenders Need: Findings and Potentials.

Q4: Utilizing high-order neighbors as external information to aid profile generation is not a novel concept. Please cite and compare with existing methods like Knowledge Plugins [5] and Language-Based User Profiles for Recommendation [6].

[5] Knowledge Plugins: Enhancing Large Language Models for Domain-Specific Recommendations.

[6] Language-Based User Profiles for Recommendation.

Q5: Since positive profiles for reinforcement learning are generated using ChatGPT, it would be beneficial to include experimental results that evaluate the oracle performance when using these positive profiles as user and item profiles.

Q6: The primary experimental focus should demonstrate that the proposed profile generation method (RecLLM) better enhances CF methods. Only comparing RecLLM to LLMRec is insufficient; comparisons with other LLM-enhanced methods like RLMRec, KAR, and AlphaRec would provide a more robust validation.

Q7: To substantiate the claim that RecLLM addresses the cold-start problem, it would be beneficial to compare its performance against state-of-the-art (SOTA) cold-start methods.

**Regarding the presentation quality**

Q9: The tuning strategy lacks a clearly defined objective function, and the masking mechanism is not accompanied by clear mathematical notation. Clarifying these would enhance the readers' understanding.

Q10: The description of the method from Lines 180–187 could be further refined for readability.

Q11: Equation 7 introduces variables x and y, which have not been previously defined. It is recommended to introduce these variables earlier or clarify their meaning in the context of the equation.

---

### Note · Authors · 2024-11-20

I have read and agree with the venue's withdrawal policy on behalf of myself and my co-authors.